# A Combined Atomic and Molecular Probe Characterization of Aromatic Hydrocarbons via PALS and ESR: Methylbenzene

**DOI:** 10.3390/ma15020462

**Published:** 2022-01-08

**Authors:** Josef Bartoš, Bożena Zgardzińska, Helena Švajdlenková, Barbara Charmas, Miroslava Lukešová, Konrad Wysogląd, Magdalena Goździuk

**Affiliations:** 1Polymer Institute of SAS, Dúbravská Cesta 9, SK-845 41 Bratislava, Slovakia; helena.svajdlenkova@savba.sk; 2Institute of Physics, Maria Curie-Sklodowska University in Lublin, Maria Curie-Sklodowskiej Sq.1, PL-20-031 Lublin, Poland; konrad1274@gmail.com (K.W.); magdalena.gozdziuk@umcs.pl (M.G.); 3Faculty of Chemistry, Institute of Chemical Sciences, Maria Curie-Sklodowska University in Lublin, Maria Curie-Sklodowska Sq.3, PL-20-031 Lublin, Poland; barbara.charmas@mail.umcs.pl; 4Institute of Macromolecular Chemistry of CAS, Heyrovského Square 2, CZ-162 06 Prague, Czech Republic; lukesova@imc.cas.cz

**Keywords:** methylbenzene or toluene, positronium, free volume, PALS, spin probe dynamics, ESR, thermodynamics, relaxation dynamics

## Abstract

A combined study of one of the simplest aromatic hydrocarbons, i.e., methylbenzene (toluene) (TOL), via the annihilation of an ortho-positronium (o-Ps) probe via positron annihilation lifetime spectroscopy (PALS) and the rotation dynamics of nitroxide spin probe 2,2,6,6-tetramethyl-piperidinyl-1-oxy (TEMPO) using electron spin resonance (ESR) over a wide temperature range, 10–300 K, is reported. The o-Ps lifetime, *τ*_3_, and the relative o-Ps intensity, I_3_, as a function of temperature exhibit changes defining several characteristic PALS temperatures in the slowly and rapidly cooled samples. Similarly, the spectral parameter of TEMPO mobility in TOL, 2*A*_zz‘_, and its correlation time, *τ*_c_, reveal several effects at a set of the characteristic ESR temperatures, which were determined and compared with the PALS results. Finally, the physical origins of the changes in free volume expansion and spin probe mobility are revealed. They are reflected in a series of the mutual coincidences between the characteristic PALS and ESR temperatures and appropriate complementary thermodynamic and dynamic techniques.

## 1. Introduction

The structural–dynamic state of organic substances is usually characterized using conventional techniques such as calorimetry and dilatometry via macroscopic quantities, such as enthalpy and volume. The structural diffraction, scattering, resonance or relaxation techniques utilizing microscopic intrinsic (internal) probes, such as static and dynamic density fluctuations, and magnetic and electric polarization, due to the respecitve dipole reorientations, are also used. Besides that, the structure and dynamics of condensed compounds can also be investigated using non-standard techniques using microscopic guest probes embedded in the organics of the studied host. The most appropriate extrinsic probes in condensed matter appear to be those entities that are as small as possible because of minimized geometric perturbation and interparticle interaction with the investigated organics. The atomic-size ortho-positronium (o-Ps) used in positron annihilation lifetime spectroscopy (PALS) or small molecular-sized stable radicals, the so-called spin probes, used in electron spin resonance (ESR), seem to be probes satisfying these requirements. The characterization potential of these non-standard techniques is connected with several problems. The first consists of the revealing of the mutual relationships between the phenomenological responses of non-standard and standard techniques. This is connected with an aspect of what additional information about the given investigated material might be provided by extrinsic probe ones with respect to the standard intrinsic techniques. The other question deals with the mutual relationships between the used non-standard techniques. What is the extent of the generality or the specificity and eventually of the synergism of the observed phenomena over a wide temperature range in given organics as a function of the used microscopic probe?

Recently, joint investigations on several organic compounds of various chemical structures and morphologies using PALS and ESR were carried out to search for their characterization potential, which can be influenced by the potential interaction of the used probe with the medium’s components, in comparison with the conventional techniques, where the mutual interactions between the medium’s particles exist only [1,2,3,4,5,6,7,8,9,10]. A series of amorphous small molecular, oligo- as well as polymeric, organic substances with different, i.e., vdW- or H-bonding, intermolecular interactions between the constituent’s entities were investigated [1,2,3,4,5,6]. The changes in reorientation dynamics of one of the smallest spin probes, i.e., 2,2,6,6-tetra-methyl-piperidinyl-1-oxy (TEMPO), are expressed by the extrema separation of the outer lines of the triplet spectra, 2*A*_zz’_, at several characteristic ESR temperatures: *T*_Xi_^slow,A^_zz_, *T*_50G_, *T*_Xi_^fast,A^_zz_, where i = 1,2,... Likewise, in PALS, characteristic temperatures in solid and liquid phases correspond to the changes in the o-Ps lifetime, *τ*_3_, and related free volume, *V*_h_,: *T*_bi_^solid^, *T*_m_^PALS^, *T*_bi_^liquid^, where i = 1,2,… The characteristic temperatures determined by the TEMPO and o-Ps probes are sometimes found to coincide, and this could suggest the same or at least similar origins of the underlying physical processes governing the aforementioned changes in spin probe mobility or free volume reflected by the o-Ps lifetime, respectively. In a few cases, via comparisons of the corresponding time scales, it was possible to identify the underlying dynamic processes responsible for these coinciding effects [3,4,5,6]. Subsequently, these joint studies on the morphologically simple organics have been extended towards crystalline aliphatic acyclic hydrocarbons, such as linear n-alkanes ranging from n-hexane (C6) up to n-nonadecane (C19) [7,8,9] and one of the aliphatic cyclic hydrocarbons, i.e., cyclohexane (CHX) [10]. Both the extrinsic microscopic probes mirror the main thermodynamic transitions, such as the solid to liquid (melting) phase transition at *T*_m_^DSC^ in both the even- and odd-number members and the solid to solid (polymorphic) phase one at *T*_ss_^DSC^ in the odd n-alkanes and CHX. However, these basic phase transitions are reflected differently by the changes in various spin probe mobility regimes. Recently, the possible origin of this apparently surprising *T*_50G_ << *T*_m_^DSC^ relationship has been related to the dynamic crossover at *T*_X_ = *T*_c_^MCT^ in the local disordered regions containing the TEMPO probes in the otherwise crystalline material [9,11].

In order to verify the aforementioned findings on aliphatic hydrocarbons, we extend this joint probe effort to aromatic cyclic hydrocarbons starting with one of the most common and simplest glass-forming aromatic compounds: methylbenzene alias toluene (TOL). Toluene consisting of small effectively internal rigid molecules, which interact through van der Waals intermolecular forces, has a crystallization tendency and exhibits an extremely fragile behavior. Such behavior is manifested by the very pronounced super-Arrhenius character of the dynamics. Its basic thermodynamic [12,13,14,15,16] and dynamic properties [17,18,19,20,21,22,23,24,25,26,27] via NMR [17,18,19,20], DS [21,22,23,24] and LS [25,26] are quite well established. The significance of TOL is emphasized by the fact that it often serves as an important small molecule model system for advanced glassy physics studies under various special physical conditions, such as the physically vapor deposited (PVD) states [28,29,30,31] and the confined (CONF) ones [32,33,34,35,36].

In contrast to extensive structural or dynamic studies on the pure and various physically modified forms of TOL using standard techniques, a relatively little number of relevant papers were reported using the aforementioned non-standard techniques with various extrinsic microscopic probes [37,38,39,40,41,42,43]. Thus, TOL was one of the first organics in which one of the specific features of the PALS technique, i.e., the quenching effect of oxygen (O_2_) on the o-Ps annihilation via o-Ps to p-Ps conversion, was discovered [37,38,39,40,41]. In the sixties, the first PALS data for TOL at RT about the o-Ps lifetime of 3.15–3.24 ns in the degassed state compared to 2.4–2.55 ns in the non-degassed (air saturated) state and about the relative o-Ps intensity with the opposite trend were obtained and described [37,38]. Only one PALS study as a function of *T* in the solid state at 88 K and over the restricted liquid state of TOL from 198 K up to 308 K was reported by Thosar et al. [40]. Earlier [37,38,39,40] and the latest [41] PALS studies on TOL via two- or three-component fits, respectively, showed almost independence of the longest annihilation component of the PALS spectra on the number of annihilation components included. On the other hand, the values of the o-Ps relative intensity exhibit rather large relative scatter among the various authors [37,38,39,40,41].

Regarding the molecular probing of TOL, a few ESR investigations were also carried out using the small and rigid spin probe 2,2,6,6-tetramethyl-4-oxopiperidinyl-1-oxy (TEMPON) over the restricted temperature range of the fast motion regime from 214 K (1.20*T*_m_) up to 250 K (1.40*T*_m_) [42], where *T*_m_ = 178 K [12,13,14]. The obtained correlation times of the spin probe TEMPON in TOL over this very narrow *T* range exhibited Arrhenius type of behavior.

The aforementioned mutual coincidences between the PALS and ESR temperatures in aliphatic hydrocarbons studied so far seem to suggest that the origins of underlying processes responsible for the changes in free volume expansion of the host medium or the guest probe reorientation in these media, respectively, are the same or at least similar. Thus, the natural specific questions in connection with TOL as a representative of aromatic hydrocarbons arise: What are evolutions of the free volume microstructure via o-Ps annihilation and of the spin probe TEMPO dynamics within the slow and fast motion regimes below or above *T*_50G_ (or *T_c_*), respectively? What are the characteristic PALS and ESR temperatures for TOL in an extraordinary wide temperature range not covered so far? And finally, what are the mutual relationships between the PALS and ESR temperatures as well as their potential relationships to various characteristic thermodynamic and dynamic temperatures as obtained using standard intrinsic techniques?

The aim of this paper is to present a combined PALS and ESR study on the host toluene medium using the respective microscopic guest probes over a wide temperature range with focus on their detailed investigation of the free volume microstructure and the guest probe dynamics in the solid phase, at the solid to liquid phase transition as well as in the liquid phase. The goal of this joint approach is to reveal and establish the characterization potential of both external probe techniques via their comparison with conventional thermodynamic and dynamic ones as well as their mutual comparison.

## 2. Materials and Methods

Methylbenzene alias toluene (TOL) with the summary formula C_7_H_8_ and the structural one C_6_H_5_-CH_3_ from Sigma & Aldrich Inc., Germany, of the purity >99.5% was used.

### 2.1. DSC

Calorimetric studies were performed using Differential Scanning Calorimetry DSC PYRIS Diamond Perkin Elmer, USA. Before the measurements the TOL samples were placed in aluminum pans. Sample mass was about 5 mg. The DSC studies of the TOL phase and dynamic transitions for all the samples were carried out from 103 K, but initial temperatures of the controlled rate of temperature change were different: 243, 223 and 203 K. The cooling rates were 1, 5 and 20 K/min, but the heating rate was +10 K/min for all the samples. The empty crucible was the standard sample. Helium was used as the curtain gas and nitrogen was used as a purging gas to avoid frost on the apparatus. Calibration for heat flux and temperature was done with indium and water. During the studies the cooling and heating thermograms were registered, which were used for determination of characteristic DSC temperatures *T*_onset_, Δ*H* and *T*_g_ and Δ*C_p_* of the ongoing processes.

### 2.2. PALS

For PALS measurements a standard fast-slow delayed coincidence spectrometer with γ detectors equipped in BaF_2_ scintillators was used. The resolution time was about 260 ps, FWHM. The total number of counts was 1.2 × 10^6^ counts in each spectrum that was collected over one hour. The liquid sample was poured into the chamber dedicated to liquid measurements, in which the 0.8 MBq of ^22^Na source in a Kapton^®^ envelope was located. PALS spectra were collected over a very wide temperature range from 10 K to 200 K and from 110 K up to 300 K, using a helium and nitrogen cooling system, respectively. Measurements were carried out under various thermal and sample state conditions, which included slow cooling (sc) vs. rapid cooling (rc), non-degassed and degassed states and their combinations. The temperature was stabilized with an accuracy of ±0.2 K. The standard freeze-pump-thaw degassing technique was used to avoid the ortho-para spin conversion by oxygen (O_2_) molecules in the degassed sample [44]. The chamber was fixed at the top of a copper rod which served as the “cold finger” of a cryostat and was heated to a set temperature. The PALS spectra were analyzed using the LifeTime (LT) 9.2 program [45] assuming three discrete components convoluted with the instrumental resolution curve. The shortest-lived component was ascribed to the decay of singlet para-Ps (here fixed at 125 ps), the intermediate one—to the annihilation of free positrons (with lifetime in the range from 286 ps to 410 ps in solid and liquid phase, respectively) and the longest-lived one—to the decay of o-Ps. The correction for annihilation in the Kapton^®^ envelope was applied (intensity: 10%, lifetime: 382 ps).

### 2.3. ESR

ESR measurements of the spin system toluene/TEMPO were performed on the X-band Bruker—ER 200 SRL (Stuttgart, Germany) spectrometer operating at 9.4 GHz with a Bruker BVT 100 temperature variation controller unit. ESR spectra of the deoxygenated TOL doped ~5 × 10^−4^ M TEMPO were recorded after cooling the degassed sample with an effective cooling rate ~−4 K/min in a heating mode over a wide temperature range from 100 K up to 250 K with steps of 5 K. After cooling down to the starting temperature, the sample was kept for a few minutes to reach the stable signal in time. To achieve the thermal equilibrium, the sample of TOL was kept at a given temperature for 10 min before the start of two spectra accumulations. The temperature stability was ±0.5 K. The microwave power and the amplitude of the field modulation were optimized to avoid signal distortion. Evaluation of the ESR spectra was performed in terms of the spectral parameter of mobility, 2*A*_zz’_, as a function of temperature with subsequent evaluation of the spectral parameter of mobility *T*_50G_ parameter [46] and further additional characteristic ESR temperatures describing the slow to fast transition zone over a more or less wide temperature interval around *T*_50G_ in detail as well as fine effects in both the slow and the fast motional regimes: *T*_Xi_
^slow,Azz^, *T*_Xi_
^fast,Azz^ [3,4,5,6]. Correlation times of the spin probe TEMPO rotation were estimated via a detailed spectral shape line analysis using the semi-empirical equations for both the fast motion regime [47,48,49] and further evaluated via detailed spectral shape simulations using the Nonlinear Squared Line (NLSL) program [50]. Their subsequent analysis provided the characteristic ESR temperatures *T*_Xi_^slow,τ^, *T_c_* and *T*_Xi_^fast,*τ*^ similarly to Refs. [4,6].

## 3. Results

### 3.1. DSC Results

The basic macroscopic characterization of TOL was made by cooling the sample with a cooling rate of −5 K/min followed by heating with a heating rate of +10 K/min over a wide temperature range from 100 K up to RT using differential scanning calorimetry (DSC). This provided the basic thermodynamic DSC temperatures connected with various phase and dynamic transitions. They include on cooling hot crystallization from liquid to partially crystalline solid phase at *T*_hc,on_^DSC^ = 143 K with Δ*H*_hc_ = −0.18 kJ/mol followed by on heating using dynamic, i.e., glass to liquid, transition (devitrification) at *T*_g,on_
^DSC^ = 119 K with subsequent additional cold crystallization, i.e., partially crystalline solid to more crystalline solid phase, transition at *T*_cc,on_^DSC^ = 135 K with Δ*H*_cc_ = −0.18 kJ/mol followed by crystalline solid to liquid (melting) phase transition at *T*_m,on_^DSC^ = 178 K—Figure 1.

In addition, another slower and faster cooling rate followed by a standard heating rate of +10 K/min were applied in an effort to mimic approximately different modes of the sample formation and measuring regime in the detailed PALS and ESR studies, as discussed in the Results section—Figure 2a,b. In general, the character of the DSC response of TOL and the related characteristic DSC temperatures depend significantly on the used cooling rate due to the medium crystallization ability of the relatively small but asymmetric TOL molecules. Thus, in the case of the very slowly cooled sample with −1 K/min, only one exothermic effect is found from the total crystallization occurring at *T*_hc,on_ (−1 K/min) = 161 K with Δ*H*_hc_ = −4.3 kJ/mol. Then, on heating the DSC scan of this very slowly cooled material only one endothermic effect is also exhibited connected with melting of the crystalline form of TOL at *T*_m,on_ = 181 K and Δ*H*_m_ = +5.6 kJ/mol. On the other hand, relatively very rapid cooling of the TOL sample with −20 K/min indicates, similarly to the basic −5 K cooling mode, two effects: a slight partial crystallization (small exotherm) followed by a step effect from vitrification one. Subsequently, on heating, three effects take place: devitrification, cold cooling and melting. Moreover, we observed that in the case of rapid cooling the starting temperature 253 K, 243 K and 223 K has no effect on the semicrystalline character of the TOL sample.

### 3.2. PALS Results

Several measuring cycles of o-Ps annihilation data collection on TOL over three different temperature ranges under various formation, i.e., cooling and sample state conditions, i.e., degassed and non-degassed state, were performed. These temperature ranges included a very low *T* region of the solid phase from 10 K to the main thermodynamic solid to liquid transition at *T*_m_^DSC^, an intermediate *T* region situated around the main thermodynamic solid to liquid phase transition around *T*_m_^DSC^ and finally a relatively high *T* region of the liquid phase above *T*_m_^DSC^ up to *RT*. The obtained o-Ps lifetime, *τ*_3_, and the relative o-Ps intensity, I_3_, results are summarized in Figure 3, Figure 4 and Figure 5.

Figure 3 displays two types of the PALS results as obtained in heating mode from the two PALS set up configurations with a helium or nitrogen cooling system.

The first type of heating run started from a very low temperature of 10 K on the deeply solidified non-degassed TOL sample up to 170 K. Such a sample marked as the so-called quasi-rapid cooled (qrc) one was prepared via a combination of rapid cooling from RT to 180 K followed by slower cooling from 180 K down to 10 K over 2 h. By extension of the usual measurement range (above 100 K) we were able to detect the first discontinuities in both the o-Ps quantities at *T*_b1_^sol,^*^τ^*^3^, *T*_b1_^sol,I3^(qrc)~115 K, which can be ascribed to the glass-to-supercooled liquid transition in the amorphous regions of partially crystalline TOL. On crossing *T*_g_^PALS^, *τ*_3_ vs. *T* dependence has a slightly increasing linear character, while a reduction in the I_3_ vs. *T* plot around *T*_b2_^I3^(qrc)~140 K is similar to the situation in the rapidly cooled (rc) sample—see below.

Another type of heating run in intermediate and high *T* regions including the solid to liquid transition range and the liquid state consists of heating from 110 K up to 300 K on the slowly cooled degassed TOL sample. Although the heating runs differ in the temperature step Δ*T* (the temperature interval between PAL spectra), being +5 K or +1 K, respectively, a step-like change in both the o-Ps annihilation quantities connected with solid to liquid state transformation at the characteristic PALS temperature *T*_b2_*^τ^*^3^ (sc) = *T*_b3_^I3^(sc) = *T*_m_^PALS^ ≅ 178 K is rather slightly dependent on the two applied Δ*T*s. The most pronounced feature in these heating runs on the slowly cooled TOL sample is the presence of the maximum in the o-Ps intensity at the characteristic PALS temperature *T*_b2_^sol,I3^ (sc) situated at around 160 K. Finally, at higher temperatures, another bend effect occurs in the o-Ps lifetime giving to rise by the characteristic PALS temperature *T*_b1_^liq,^*^τ^*^3^ = 225 K. This effect has no equivalent in the relative o-Ps intensity I_3_, having an approximately constant value of 44%.

Figure 4 presents the PALS data from a heating run over the intermediate *T* range from 110 K to 195 K on the TOL samples with different formation histories. At the lowest *T*s we observe the slightly higher *τ*_3_ values for the rapidly cooled non-degassed (rc-TOL) sample compared to the slowly cooled degassed (sc-TOL) one. Next, a strong gradual growth in sub-*T*_m_^PALS^ region with an onset at ca. *T*_b3,on_^sol,^*^τ^*^3^(rc)~165 K is in contrast to a rather sharper one for the sc-TOL at *T*_b3_^sol,^*^τ^*^3^(sc) = 178 K. One of the most important findings concerns the I_3_ quantity. At the lowest *T*s one finds the same I_3_ values followed at middle *T*s by the dramatic difference in the I_3_ vs. *T* course with the maximum at *T*_b2_^sol,I3^(rc)~135 K in rc-TOL in contrast to *T*_b2_^sol,I3^(sc)~160 K in sc-TOL. Note that the former value is rather close to the previous *T*_b2_^I3^(qrc)~140 K from the heating scan on the quasi-rapid cooled TOL sample prepared initially via very rapid cooling of TOL from RT to 180 K followed by further cooling down to 10 K. Next, above *T*_m_^PALS^ both the *τ*_3_−*T* plots are finished by rather slightly lower values for the non-degassed TOL, while the I_3_−*T* plots exhibit quite different courses due to the presence of the oxygen in the TOL sample. Thus, the behavior of *τ*_3_ at lower *T*s in the solid phase contrary to the general expectation [37,38,39,40,41,44] indicates rather the dominancy of the formation history factor, i.e., the method of sample preparation, over the o-Ps to p-Ps conversion effect of O_2_ in the PALS response in the solid state, whereas in the liquid state, the presence of oxygen leads to more significant changes in *τ*_3_ and especially in I_3_ compared to the degassed situation.

Finally, Figure 5 displays the PALS results from measurements in a cooling mode on the degassed TOL sample with different temperature steps, Δ*T* = −10, −5, −2 and −1 K, and on the non-degassed TOL sample with temperature step Δ*T* = −10 K. In the case of the degassed TOL sample, a gradual slow cooling in the liquid state from RT down to ca. 125 K, the *τ*_3_ vs. *T* plot, at relatively higher *T*s exhibits two linear regions defining the characteristic PALS temperature *T*_b1_^liq,^*^τ^*^3^~225 K similarly to the heating run in Figure 3. Two approximately linear regions are also evident in the I_3_−*T* one, but at the essentially lower characteristic PALS temperature *T*_b1_^liq,I3^~180 K lying in the vicinity of the *T*_m,on_^DSC^. The most important finding from these cooling runs is that the end values of the higher *τ*_3_ and I_3_ quantities defining the solidification (freezing) temperature, *T*_sol_, as seen with PALS depend on the temperature step Δ*T*. Thus, the higher the Δ*T* step, the lower the characteristic PALS temperature, *T*_sol_, of the step-like change of both the o-Ps annihilation quantities, which represents a phase transformation of TOL via solidification, predominantly via crystallization. In particular, the characteristic PALS temperatures of solidification in terms of *τ*_3_ are *T*_sol_*^τ^*^3^(Δ*T*) = 163, 162 and 159 K for Δ*T* = −1, −2 or −5 and −10 K, respectively. On the other hand, a remarkable finding is the superposability of the relative o-Ps intensity, with the maximum at *T*_b1_^sol,I3^ ≅ 158 K in the solid state. Note that this value lies in the vicinity of the I_3_ maximum for the slowly cooled TOL sample measured in a heating mode in Figure 3.

In the case of the non-degassed TOL sample, the *τ*_3_ values are slightly lower than for the degassed ones at low *T*s, but differ significantly from those in the higher *T* range with the signature of two regions with similar *T*_b1_^liq,^*^τ^*^3^ as in the previous degassed one. The *τ*_3_ values at RT for both the non-degassed and degassed TOL samples are dramatically distinct due to the p-Ps to o-Ps conversion via the dissolved oxygen in the former medium, which is consistent with the literature data [37,38,39,40,41,44], while the temperature dependence of both the o-Ps quantities for the former over a wide T range is new information. Interestingly, at and below the phase transition the o-Ps lifetimes are rather insensitive to the presence or the absence of O_2_ in strong contrast to the relative o-Ps intensities due to radiation-chemistry and slow dynamics of the TOL molecules. In partial summary, the PALS measurements with a set of different cooling steps show that the TOL medium can be relatively easily undercooled up to 20 K below *T*_m_^DSC^ during even relatively slow cooling and further that the applied Δ*T* change affects rather significantly the solidification (liquid to solid) phase transition.

### 3.3. ESR Results

Figure 6 shows the spectral evolution of the ESR spectra of the spin probe TEMPO in TOL as a function of temperature over the whole measured temperature range from 100 K up to 250 K. Three basic regions of the distinct character of the ESR spectra are evident. At the lowest as well as at the highest temperatures, regions A and C, the triplet spectra, have a monomodal character, while in the intermediate temperature region B in between ca. 155–175 K they exhibit the bimodal form. In the first and the third region the spin probe TEMPO moves in the true slow or fast motion regime, respectively, while the second one is characterized by the co-existence of both the slow and fast moving guest molecules, which reflects their two different surroundings.

Figure 7 displays the extrema separation of the outermost lines of the triplet ESR spectra, 2*A*_zz’_, as one measure of the spin probe TEMPO mobility in TOL as a function of temperature *T* over a wide temperature interval from 100 K up to 250 K. The typical sigmoidal course of 2*A*_zz’_ exhibits two basic regions of distinct motion regimes of the spin probe TEMPO in the TOL medium. One, at low temperatures with the relatively high values of 2*A*_zz’_ around 67.5 Gauss, is the so-called slow motion regime and the other, at higher temperatures with the relatively low 2*A*_zz’_ s with the relatively low values around 33 Gauss, is the so-called fast motion regime. Conventionally, this slow to fast motion regime transition defines the most pronounced characteristic ESR temperature *T*_50G_ = 177.5 K as the temperature at which the extrema separation occurs of the outer lines of the triplet signal corresponding to the correlation time scale of spin probe reorientation *τ*_c_~2 −4 x10^-9^ s [6,46]. In addition to this main effect in the 2*A*_zz’_ vs. *T* dependence, the former motion region shows up some fine structure as demonstrated in insets in Figure 7. Here, three distinct zones of different thermal behavior of 2*A*_zz’_ can be distinguished, which define further two characteristic ESR temperatures: *T*_X1_^slow,Azz^ = 115 K and *T*_X2_^slow,Azz^ ≅ 135 K. The first region at the lowest temperatures from the starting temperature up to *T*_X1_^slow,Azz^ = 115 K with the constant 2*A*_zz’_ value of 67.5 Gauss represents the effectively immobilized spin probe TEMPO in the rigid TOL matrix. The second zone from *T*_X1_^slow,Azz^ up to *T*_X2_^slow,Azz^ ≅ 135 K is characteristic due to a slightly decreasing 2*A*_zz’_ trend related to the partially motionally averaged anisotropy of the hyperfine splitting *A* tensor of the slowly moving probes—see the left inset in Figure 7. Next, after crossing *T*_X2_^slow,Azz^ the further partial averaging of the anisotropy of the TEMPO molecules in TOL occurs due to their increased spin probe TEMPO mobility in the TOL matrix. The first indication of the narrow spectral component in the complex spectra seems to occur at around *T*_ini_^fast,Azz^~160 K (Figure 7) leading to the appearance of the fast component with 2*A*_zz’_~35 G in Figure 6. Starting at ca. *T*_X3_^slow,Azz^ = 175 K, the slow to fast motional regime transition occurs over a very narrow temperature range of ca. 10 K around *T*_50G_ = 177.5 K followed by *T*_X1_^fast,Azz^ = 180 K by the pure fast motion regime with the motion narrowed triplet spectra with a slightly decreasing trend in the 2*A*_zz’_ values. Finally, a small step effect in the 2*A*_zz’_ vs. *T* plot at around 215 K defining the crossover ESR temperature *T*_X2_^fast,Azz^ is found—see the right inset in Figure 6.

Figure 8 shows the typical spectral simulation results and Figure 9 presents the correlation times, *τ*_c_, as another measure of the rotation dynamics of the spin probe TEMPO in TOL as a function of the inverse temperature, 1/*T*, together with the temperature dependence of the relative fraction of the broad or/and narrow spectral component *F*_slow_, *F*_fast_ = 1−*F*_slow_ over a wide temperature interval from 100 K up to 250 K. Three basic regions of distinct behavior of *τ_c_* as well as of Fs are found: (i) low-*T* region A, (ii) intermediate-*T* region B and finally (iii) high-*T* region C. The lowest temperature zone A with the broad monomodal spectra corresponding to the slow moving spin probes of TEMPO provide the relatively long and slightly decreasing correlation time values corresponding to the restricted mobility. Above 145 K, i.e., in intermediate-*T* region B, marked by *T*_X1_^slow,τ^ = *T*_ini_^fast,^*^τ^* = 150 K, both slow and fast components are superimposed indicating the co-existing spin probe TEMPO populations in distinct motional regimes up to *T_c_* = 180 K. The former component accelerates significantly above ca. *T*_X2_^slow,^*^τ^*~157 K in contrast to the latter, which changes slightly with increasing temperature. Finally, at the highest temperatures (region C) the narrow spectra from the fast moving spin probes of TEMPO dominate the other monomodal motion regime with two distinguished sub-regimes, giving the characteristic ESR temperature *T*_X_^fast,^*^τ^* = 218 K.

## 4. Discussion

### 4.1. The Mutual Relationships between the PALS and ESR Data

The PALS and ESR responses measured in the heating mode in Figure 3, Figure 4, Figure 7 and Figure 9 exhibit several effects at the respective PALS and ESR temperatures, which reflect the underlying changes in the structural or/and dynamic state of the TOL medium. In our search for the origin of these changes in the free volume structure or/and the guest probe dynamics it is useful first to compare their eventual coincidences reflected in the corresponding characteristic PALS and ESR temperatures. On going from below, we can observe the following set of five coincidences:*T*_X1_^slow,Azz^~115 K~*T*_b1_^sol, *τ*3,I3~^115 K(1)
*T*_X2_^slow,Azz^~135 K ≅ *T*_b2_^sol,I3^(rc)~137 K (2)
*T*_X1_^slow,*τ*^ = *T*_X,ini_^fast,*τ*^~150 K ≈ *T*_b3_^sol,*τ*3^(rc)_~_*T*_b1_^sol,I3^(sc)~160 K(3)
*T*_50G_~177.5 K ≅ *T_c_* =180 K~*T*_b3_*^τ^*^3,I3^(sc)(4)
*T*_X2_^fast,Azz,^~215 K ≅ *T*_X2_^fast,*τ*^~218 K~*T*_b1_^liq,*τ*3~^225 K (5)

The most pronounced effects in both the PALS and ESR responses are given by the fast to slow regime transition of TEMPO in the TOL medium at the conventional ESR temperature *T*_50G_~177.5 K, or rather, more precisely, at the true fast temperature *T_c_* = 180 K, which is close to the sharp stepwise increase in both the o-Ps annihilation parameters, *τ*_3_ and I_3_, at around 178 K in the slowly cooled sc-TOL sample—Equation (4). In addition, some further coincidences also exist between the less pronounced effects below and above this temperature. Thus, the first decrease in 2*A*_zz‘_ within the slow regime at *T*_X1_^slow,Azz^~115 K, approximately coinciding with the first increase in *τ*_3_ and I_3_ at *T*_b1_^sol,*τ*3,I3^ ~110–115 K, is found—Equation (1). The next slight decrease in 2*A*_zz‘_ within the slow regime at *T*_X2_^slow,Azz^~135 K lies in the vicinity of the maximum in I_3_ for the rapidly cooled rc-TOL sample at *T*_b2_^sol,I3^(rc)~137 K—Equation (2). On the other hand, from spectral simulations in Figure 9, an onset of the fast component within the superimposed slow-fast regime at *T*_X,ini_^fast,*τ*^ = 150 K is close rather to the I_3_ maximum for the slowly cooled sc-TOL sample at *T*_b1_^sol,I3^(sc)~160 K—Equation (3). Finally, the slight decrease in 2*A*_zz‘_ within the fast regime at *T*_X2_^fast,Azz^~215 K and the crossover between two fast sub-regimes at *T*_X_^fast,*τ*^~218 K coincide plausibly with *T*_b1_^lq,*τ*3^~225 K—Equation (5) from Figure 3 and Figure 5. In all these cases, the observed effects are due to the onset of some local cooperative or collective dynamics, which can be discussed via their comparison with thermodynamic and dynamic properties.

### 4.2. Thermodynamic Interpretation of the PALS and ESR Data

First, in order to identify the underlying processes, we compare the revealed mutual coincidences between the PALS and ESR responses with the thermodynamic transitions due to either various transitions within a given thermodynamic phase or between different thermodynamic ones at phase transitions. From confrontation of the basic DSC thermogram for −5 K/min in Figure 1 and for two cooling regimes of −1 K/min and −20 K/min in Figure 2 with the relevant Figure 3, Figure 4, Figure 7 and Figure 9, we arrive at the following three mutual relations:
*T*_X1_^slow,Azz^~115 K~*T*_b1_^sol,τ3,I3~^115 K ↔ *T*_g,on_^DSC^ = 117.5 K(6)
*T*_X2_^slow,Azz^~135 K = *T*_b2_^sol,I3^(rc)~137 K ↔ *T*_cc,on_^DSC^ = 137 K(7)
*T*_50G_ = 177.5 K ≅ *T_c_* = 180 K~*T*_m_*^τ^*^3,I3^(sc)~178 K ↔ *T*_m,on_^DSC^ = 178 K(8)

Starting in the lowest temperature region, the first decrease in 2*A*_zz’_ at *T*_X1_^slow,Azz^ coinciding approximately with *T*_b1_^sol,*τ*3,I3^ correlates quite plausibly with devitrification at *T*_g,*i*_^DSC^. Thus, the first partial averaging of the magnetic anisotropy of the spin probe TEMPO in TOL and the first expansion in the local free volume are related to the glass to supercooled liquid transition in the amorphous regions/phase of the TOL medium. Since the TOL sample has a partially crystalline character, some additional phase transformation, the so-called cold crystallization, at *T*_cc,*i*_^DSC^ can occur during its heating, which subsequently can also contributes to further partial averaging of the magnetic anisotropy of the spin probe TEMPO in TOL via the second decrease in 2*A*_zz’_ at *T*_X2_^slow,Azz^ coinciding approximately with a decrease in the I_3_ intensity at *T*_b2_^solid,I3^ in the rc-TOL sample. The increased mobility of the TOL molecules caused by the phase transformation resulting in the two-phase medium leads to an occurance of dynamic heterogeneity in the spin probe reorientation, as seen from the NLSL simulations in Figure 8 and Figure 9. Finally, on further increase of the temperature, the most pronounced effects in both the ESR and PALS responses fall into the main solid to liquid phase transformation at *T*_m,on_^DSC^, so that the corresponding changes in spin probe dynamics and free volume expansion are related to a collective phase transition phenomenon.

### 4.3. Dynamic Interpretation of the PALS and ESR Data

In the previous section, some of the five coincidences between effects in the PALS and ESR responses were explained by considering the thermodynamic data accessible due to DSC measurements. However, the phenomenology of external probing techniques is essentially richer than the DSC one, so that these thermodynamics alone are not sufficient to explain all of the found effects and their coincidences. Thus, some of the remaining ones require another, dynamic interpretation approach. This can be performed via comparison at (i) a level of the characteristic PALS and ESR temperatures vs. some characteristic dynamic (DYN) ones and, in the case of the ESR data, at (ii) a level of the time scales of spin probe rotation in a given medium vs. those of various relaxation modes of the medium.

Figure 10 represents a compilation of all the present dynamic data for amorphous TOL. They include those from conventional direct measuring techniques, such as viscosimetry (VISC) [51,52,53], light scattering (LS) [25,26], nuclear magnetic resonance (NMR) [17,18,19,20] and dielectric spectroscopy (DS) [21,22,23,24], over an extremely wide temperature range including the normal liquid and supercooled liquid states of TOL from 380 K to *T*_g_ as well as the glassy one of TOL from *T*_g_ down to 70 K. It is evident that TOL can be relatively easily supercooled with a relatively strong slowing down of the primary relaxation dynamics. On going from above, the primary *α* relaxation time, *τ_α_*, exhibits the typical Arrhenius behavior down to *T*_A_ = 225 K [54] or 220 K [55], which is marked by the Arrhenius temperature as one of the basic characteristic dynamic temperatures in the normal liquid state of matter. On further decrease of the temperature, the super-Arrhenius behavior with a very sharp growth of the *τ_α_* values with decreasing temperature indicates that TOL is a highly fragile glass former [22,23]. In addition, TOL, as a typical representative of Type B glass formers [24], exhibits a pronounced secondary *β* process with typical Arrhenius behavior not only in the glassy state, but also above *T*_g_^DSC^ in the strongly supercooled liquid state.

Returning to the PALS and ESR data on TOL, we find that the remaining characteristic PALS and ESR temperatures can be related to some dynamic features found from the relaxation map of TOL in Figure 10 and/or from appropriate treatments of the structural relaxation times in terms of relevant phenomenological models. 

Thus, the characteristic PALS temperature *T*_b1_^liq,*τ*3^~225 K mentioned in Figure 3 and Figure 5 in Section 3.2 is in good agreement with the Arrhenius temperature *T*_A_ = 225 K [54] or 220 K [55] as one of the basic characteristic temperatures of the structural dynamics in all organic glass formers, which for TOL lies about 47 or 42 K above *T*_m_^DSC^, i.e., ca.(1.25 ± 0.01)*T*_m_^DSC^. It is well known that a relatively slighter change in the *τ*_3_ vs. T dependence in many amorphous and crystallizing organics at the relatively high *T* starting above, at and even somewhat below the corresponding melting temperature, *T*_m_, is observed [56,57,58,59,60,61,62]. This general trend is explained in several ways, such as (i) the digging of holes in liquid medium by o-Ps itself [61], (ii) the closeness of the time scale of medium dynamics to the o-Ps lifetime [58,59,60] or (iii) the formation of a “bubble” o-Ps state in a low viscosity medium [62,63]. Within the last hypothesis (iv) a reduction in the “bubble” volume expansivity in TOL on crossing *T*_b1_^liq,*τ*3^ during cooling (or heating) appears to be closely related to a crossover in the structural dynamics from the Arrhenius character above *T*_A_ to the super-Arrhenius course below *T*_A_. Thus, the “bubble” behavior seems to be sensitive to the change in dynamics of TOL.

As for the characteristic ESR temperature *T*_X2_^fast,Azz^ ≅ *T*_X_^fast,*τ*^ = 215–218 K lying relatively close to *T*_A_ = 220 K [55], this finding indicates that the slowing down of the spin probe TEMPO reorientation in TOL on cooling within its fast motion regime is also related to the Arrhenius to super-Arrhenius crossover of the dynamics in TOL. This is also suported by the empirical fact found by some of us for many other amorphous organics that at the characteristic ESR temperature *T*_X_^fast^, at which the o-Ps lifetime corresponding to the local spherical free volume, *V_h_^sphere^*, equals the van der Waals volume of TEMPO, *V_TEMPO_^W^*, *τ*_3_(*V_h_^sphere^* = *V_TEMPO_^W^*) = 2.72 ns, a change in the dynamics occurs [4,7,8,9]. The o-Ps lifetime measured in TOL at 218 K *τ*_3_ = 2.75(8) ns and confirms the above. In other words, a transition from the full to partial motional averaging of the magnetic anisotropy or the deceleration of rotation dynamics is related to the high-T dynamic crossover of TOL due to the onset of restricted local free space conditions on cooling the TOL sample.

One of the most interesting findings of this joint PALS and ESR work is the marked distinction in the respective behavior of both o-Ps annihilation quantities in the rapidly cooled non-degassed TOL sample vs. the slowly cooled degassed one during their PALS measurement in heating mode—Figure 3. Both the o-Ps lifetime, *τ*_3_, and especially the relative o-Ps intensity, I_3_, exhibit a dramatic difference over the whole temperature range studied, i.e., 120 K−190 K. These consist in an earlier onset of the growth in *τ*_3_ at ca. *T*_b3,on_^sol,*τ*3~^160 K for TOL(rc) compared to that for TOL(sc) at ~180 K. At the same time, this is accompanied by a large shift in the characteristic PALS temperatures of the relative o-Ps lifetime intensity maximum at *T*_b2_^sol,I3^(rc) = 137 K < *T*_b2_^sol,I3^(sc) = 160 K. It is generally accepted that while the former quantity in the absence of O_2_ is determined mainly by the free volume geometry (size, shape), the latter, as a measure of the o-Ps formation probability, depends not only on the free volume population, but also on various radiation-chemical processes, which occur under the given structural-dynamic conditions of medium determined by internal variables, such as the chemical structure of the constituent’s molecules and the absence or the presence of O_2_, as well as by external ones, such as temperature and pressure [37,38,40,44]. Concerning the o-Ps lifetime, both the presence or the absence of oxygen and the used formation histories of preparation of the TOL sample seem to be of slight relevance because of rather weak dependence of the *τ*_3_ values and the closely related geometry of local free volume entities in the solid TOL medium. On the other hand, the I_3_ vs. *T* plot for the rapidly cooled TOL sample shows first an increase with a maximum at the relatively low characteristic PALS temperature *T*_b2_^sol,I3^(rc) = 137 K followed by a decrease down to 175 K close to *T*_m_^DSC^ = *T*_b3_^sol,*τ*3^ = *T*_m_^PALS^. This characteristic PALS temperature, which is close to the characteristic ESR one, *T*_X2_^slow,Azz^ of the second motional averaging of the magnetic anisotropy within the slow motion regime, can be related to the onset of the liquid to solid (cold) crystallization occurring in the originally less ordered TOL sample at *T*_cc,on_^DSC^. At the same time, from the relaxation map of TOL in Figure 10 it follows that the local secondary *β* process as detectable via DS takes place even above *T*_g_^DSC^. Subsequently, a short linear extrapolation of the secondary *β* relaxation time from the glassy state toward the primary α relaxation time in the supercooled liquid state provides the estimated *αβ* merging temperature: *T*_αβ_^DYN^~140 K [22]. This coincidence between *T*_cc,on_^DSC^ ≡ *T*_αβ_^DS^ suggests a close connection between the local dynamics in the amorphous phase and the phase transformation in the supercooled liquid state with their potential impact on both atomic and molecular probes behaviors. Moreover, good coincidences between the characteristic PALS and ESR temperatures with the thermodynamic *T*_cc,on_^DSC^ and dynamic *T*_αβ_^DS^ temperatures imply the presence of certain local amorphous regions in the otherwise mostly ordered material. Consequently, this altered structural-dynamics situation may, at least partly, be responsible for the partial reduction of the free volume population in the TOL sample during its heating due to the onset of cold crystallization in the rapidly cooled, originally slightly partially crystalline TOL material.

On the other hand, on heating the relatively more ordered (crystallized) TOL sample formed by slow cooling, I_3_ shifts to a bit more intense maximum at the higher characteristic PALS temperature *T*_b2_^sol,I3^(sc)~160 K followed by a decrease down to 175 K close to *T*_m,on_^DSC^ = *T*_m_^PALS^. Interestingly and apparently consistently, the maximum temperature agrees well with the initial temperature of the fast motion below, *T*_X,ini_^fast,Azz^, lying ca. 20 K below the main slow to fast motion regime transition of TEMPO at *T*_50G_, which is not too distant from the finding of *T*_X,ini_^fast,*τ*^~150 K from the simulation data—see Figure 9. These PALS and ESR findings and their mutual coincidences can plausibly be interpreted via utilization of relevant model treatments of the dynamics in amorphous TOL from Figure 10. Figure 11 displays fits of the time scales of the structural *α* relaxation in the amorphous phase of TOL in terms of two models, namely, the empirical power law (PL) equation [64], which is explained in terms of the idealized mode coupling theory (MCT) of glass transiton [65,66,67,68], and the two order parameter (TOP) model of supercooled and normal liquid state dynamics [69,70,71,72,73,74,75]. The first model is representended by the following equation:(9)$$\tau_{\alpha }\left(T\right)=\tau_{\infty ,\alpha }{\left[\frac{\left(T-T_{x}\right)}{T_{x}}\right]}^{-\mu }$$
where *τ*_∞,*α*_ is the pre-exponential factor, *T_X_* is the characteristic dynamic PL temperature or the so-called critical MCT temperature *T_c_^MCT^* and *µ* is a non-universal coefficient. It is argued that the PL equation is valid for a large number of organic molecular glass formers over a rather higher temperature range [64,65,66,67]. As it is known, the MCT works very well also for the relatively lower viscosity regime [68]. In reality, although the viscosity does not diverge at *T_X_* = *T_c_^MCT^*, several analyses of the slightly supercooled and normal liquid dynamics in various organic glass formers in terms of the extended mode coupling theory (E-MCT), removing this singularity, provide the same crossover temperature in the supercooled liquid phase. Thus, the *T_X_* parameter marks two distinct regimes of the weak and strong supercooled liquid dynamics at relatively higher or lower temperatures, respectively [68].

The other one is described by the following equation:(10)$$\tau_{\alpha }\left(T\right)=\tau_{\infty ,\alpha }\exp\left[\frac{E^{*}}{RT}\right]\exp\left[\frac{B^{TOP}f\left(T\right)}{T-T_{0}^{TOP}}\right]$$
where *τ_α_(T)* is the relaxation time, *τ*_∞,*α*_ is the pre-exponential factor, *E*^*^ is the activation energy above *T*_A_ ≥ T_m_, *T*_0_^TOP^ is the divergence temperature, *B^TOP^* is the coefficient and a probability function *f(T)* for the solid-like amorphous domains is given:(11)$$f\left(T\right)=\frac{1}{\exp[\kappa \left(T-T_{m}^{c}\right)]+1}$$
where *κ* reflects the sharpness of *f(T)* and *T_m_^c^* is the characteristic TOP temperature at which the free energy of crystallizing or non-crystallizing liquids is equal to the solid state energy ∆*G_lq_* = ∆*G_sol_*. The TOP model provides the physical picture of any glass-forming compound from the glassy state over supercooled liquid to normal liquid state in terms of the temperature dependence of the solid-like amorphous domain probability function *f(T)* and various physical observations might be discussed within it. This picture seems to be supported by the experimental finding from the Brillouin–Rayleigh (BR) spectroscopy in Ref. [54].

From comparison of the characteristic MCT and TOP temperatures *T_c_^MCT^* = 153 K and *T_m_^c,TOP^* = 156 K with the characteristic PALS temperature *T*_b2_^sol,*τ*3^(rc) ≅ *T*_b2_^sol,I3^(sc) = 160 K and the characteristic ESR one, *T*_X2_^slow,^*^τ^* = 157 K, it follows their quite favourable agreement. These suggest that the observed effects at these characteristic PALS and ESR temperatures appear to reflect the dynamic crossover in mobility within the local amorphous phase of the partially crystalline TOL regions.

In addition to discussion of the PALS and ESR responses in terms of the corresponding characteristic temperatures, the ESR data can also be confronted with various dynamic ones at a level of time scale. Figure 12 compares the correlation time of the spin probe TEMPO reorientation in TOL from the NLSL simulations with the characteristic time scales of both the primary *α*— and the secondary *β* processes in the amorphous TOL medium compiled in Figure 10. Two basically different regions of the mutual time scale relationship can be observed. In the normal liquid state of TOL at high temperatures above the melting point *T*_m_^DSC^, both the time scales are in plausible agreement, indicating a significant mutual coupling between the spin probe TEMPO reorientation and the rotational reorientation of the TOL molecules connected with the structural relaxation of the TOL medium. Such a relationship was recently observed by some of us also for other small molecular organics, such as propanol (PrOH) [4] and glycerol (GL) [5]. On the other hand, at lower *T*s a dramatic decoupling between the spin probe dynamics and the medium molecule one in TOL occurs similarly as for PrOH and GL. However, one essential difference associated with the main transition at *T_c_* between the true fast motion regime of TEMPO and the fast component in the dynamic heterogeneity region of TEMPO in PrOH [4] and TOL does exist. In the former case of a fully amorphous compound this transition proceeds in a non-stepwise fashion, while in our latter one a pronounced stepwise effect appears due to the phase transformation between the amorphous phase and the partially crystalline solid TOL at *T*_m_^DSC^. Thus, this difference is related to the two-phase state of the spin system of TEMPO/TOL vs. the one-phase state of the amorphous TEMPO/PrOH system. The two-phase character of the spin system TEMPO/TOL below *T*_m_^DSC^ and the influence of amorphous medium dynamics in the local amorphous region with the TEMPO probe is supported by the co-existence of the fast and slow components in a region of the spin probe dynamic heterogeneity above *T*_X,ini_^fast,*τ*^, slightly above the cold crystallization temperature, *T*_cc,on_^DSC^. Finally, in the true slow motion region, the very low activation parameters of the TEMPO dynamics indicate some small-scale dynamics of the constituent’s molecules, probably of the libration character [18,19], in local amorphous surroundings of the spin probe TEMPO.

## 5. Conclusions

A comprehensive structural-dynamic study of methylbenzene as one of the simplest organic compounds interacting via dispersion intermolecular forces using positron annihilation lifetime spectroscopy (PALS) and electron spin resonance (ESR) with two microscopic probes was performed. In general, the extrinsic PALS and ESR probe techniques provide a richer phenomenology in comparison with a series of standard characterization techniques, such as thermodynamic DSC or/and dynamic DS, NMR, LS and VISC ones. Moreover, numerous mutual coincidences between the annihilation parameters *τ*_3_ and I_3_ of the atomic o-Ps probe and rotation parameters 2*A*_zz‘_, *τ_c_^i^* and *F_i_* of the molecular spin probe TEMPO were found. While some of these mutual coincidences can be related to the macroscopic phase and state behavior as obtained via DSC, others can be explained using the relevant dynamic data. On the other hand, the latter intrinsic techniques together with appropriate models of the liquid state dynamics enable enlightment of the physical origins of the numerous mutually coinciding effects in the PALS and ESR responses. Thus, the synergic action of both the extrinsic and intrinsic techniques contributes to deeper understanding of the complex structural-dynamic situation in organic compounds over a wide temperature range including various phase and physical states of matter as detected by the former ones.

## Figures and Tables

**Figure 1 materials-15-00462-f001:**
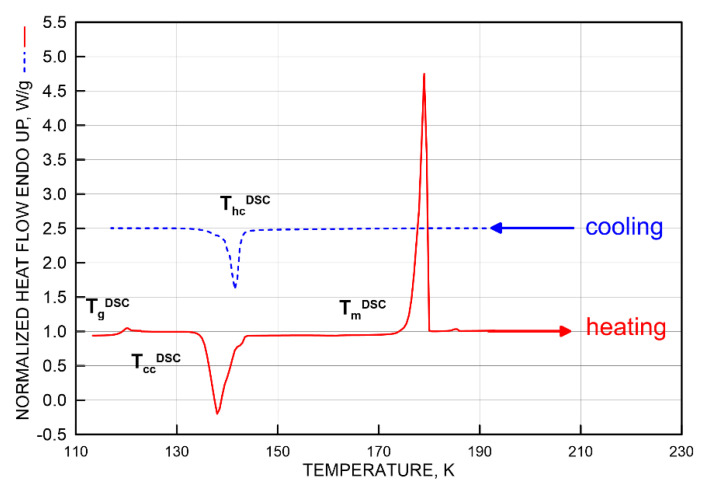
Basic DSC thermogram of TOL consisting of cooling scan followed by heating one with cooling/heating rates of −5/+10 K/min. All the exothermic effects from hot and cold crystallizations and the endothermic effects from devitrification and melting are marked by the following four onset characteristic DSC temperatures *T*_hc,on_^DSC^ = 143 K, *T*_g,on_^DSC^ = 117.5 K, *T*_cc,on_^DSC^ = 135 K and *T*_m,on_^DSC^ = 178 K as described in the text.

**Figure 2 materials-15-00462-f002:**
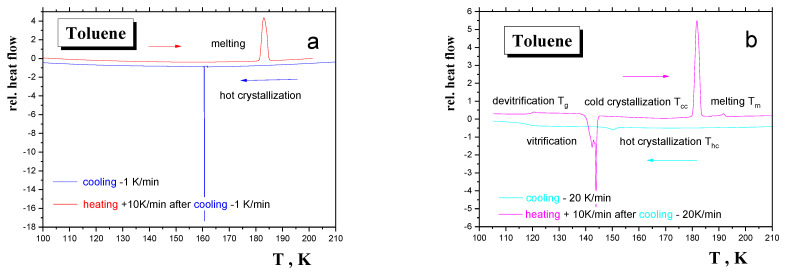
DSC thermograms for the TOL samples prepared using two distinct “boundary” formation methods: (**a**) under very slow cooling with cooling rate of −1 K/min and (**b**) under relative rapid cooling with cooling rate of −20 K/min followed by heating with heating rate of +10 K/min together with the observed thermal events.

**Figure 3 materials-15-00462-f003:**
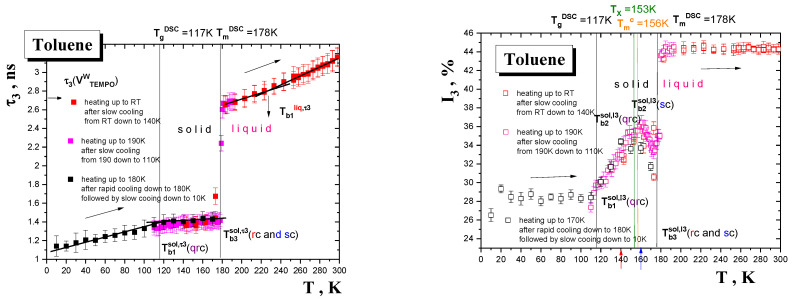
o-Ps annihilation parameters of TOL during various heating runs: (i) heating from 10 K up to 170 K after cooling from *RT* down to 10 K in 2 h consisting of rapid cooling down to 180 K, i.e., slightly above *T*_m_^DSC^ = 178 K, followed by slower cooling in the non-degassed state (black) down to 10 K, (ii) heating from 110 K up to 190 K with 2 and 1 K step after slow cooling (sc) (magenta) with Δ*T* = 1 K and (iii) heating from 140 K up to RT K with 5 K and then with 10 K step in the degassed state (red). The characteristic PALS temperatures of various effects are mentioned in the text and the characteristic DSC temperatures *T*_g,on_^DSC^ and *T*_m,on_^DSC^ from Figure 1 together with the characteristic dynamic temperatures *T*_X_, *T*_m_^c^ are also included.

**Figure 4 materials-15-00462-f004:**
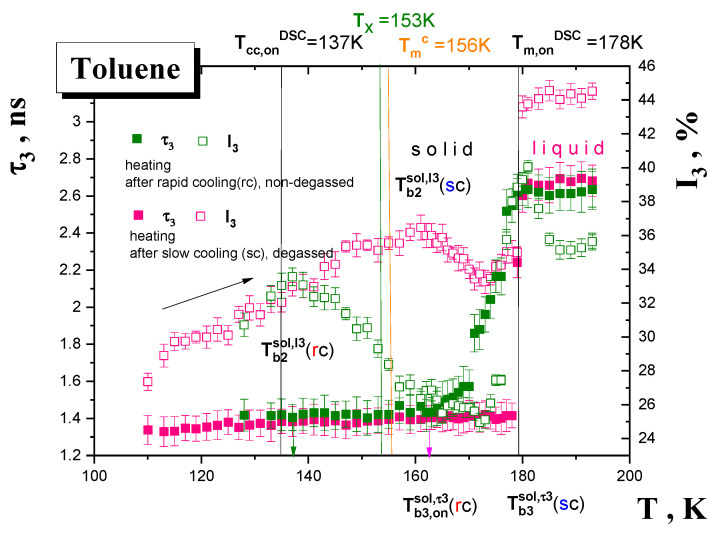
o-Ps annihilation parameters of TOL obtained during heating runs: heating from 110 K to 195 K after slow cooling (sc) in the degassed state (pink); heating from 128 K to195 K after rapid cooling (rc) in the non-degassed state (olive). The characteristic PALS temperatures are mentioned and discussed in the text. The basic thermodynamic temperatures from cold crystallization at *T*_cc,on_^DSC^ and melting at *T*_m,on_^DSC^ together with the characteristic dynamic temperatures of the two structural relaxation models (*T*_X_,*T*_m_^c^)—see Section 4—are also included.

**Figure 5 materials-15-00462-f005:**
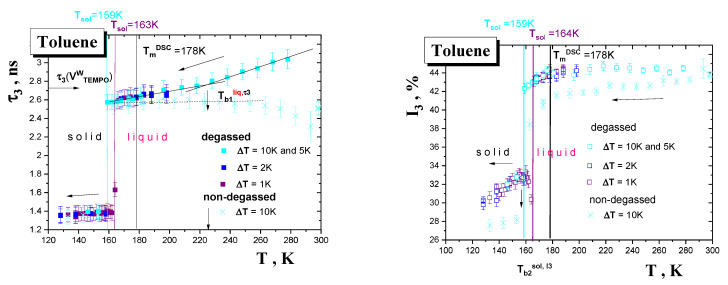
o-Ps annihilation parameters for TOL during various cooling runs on the degassed TOL sample: (i) cooling from RT with Δ*T* = 10K down to 180 K close to *T*_m,on_^DSC^ = 178 K and then with Δ*T* = 5 K down to 145 K (light blue squares); (ii) cooling from 200 K with Δ*T* = 5 K down to125 K (blue squares); (iii) cooling from 190 K with Δ*T* =1 K down to 135 K (violet squares) on the non-degassed TOL; iv) cooling from RT, first with Δ*T* = 10K down to 130 K (light blue crosses). The characteristic PALS temperature of the change in the liquid state *T*_b1_^liq,^*^τ^*^3^, of solidification *T*_sol_(Δ*T*) and the melting point at *T*_m,on_^DSC^ from DSC and finally of the change in the solid state *T*_b2_^sol,I3^ ≅ 158 K are marked in the upper or lower *T* scales, respectively.

**Figure 6 materials-15-00462-f006:**
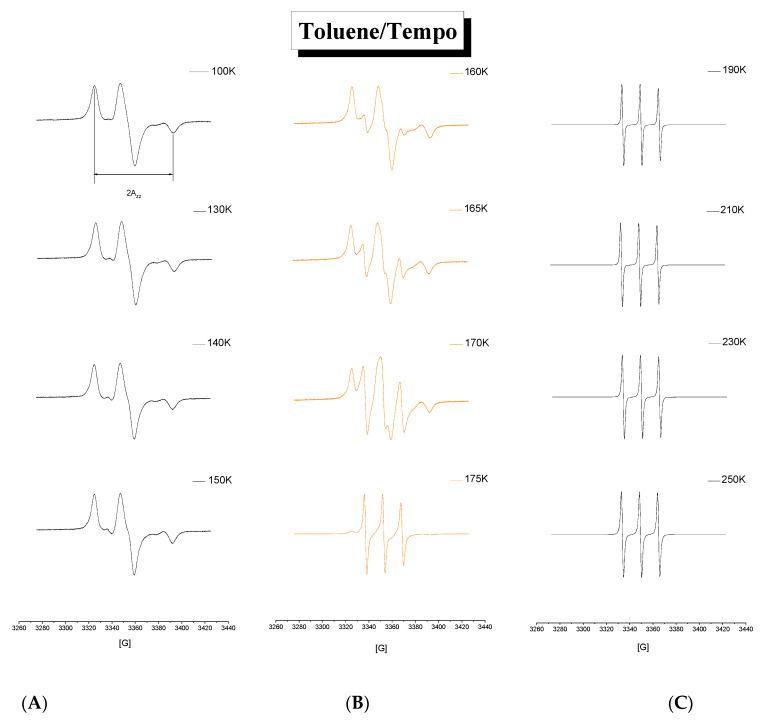
Spectral evolution of the spin TOL/TEMPO system as a function of temperature. Three main regions of the two monomodal (**A**,**C**) and the one bimodal (**B**) type of spectra are evident.

**Figure 7 materials-15-00462-f007:**
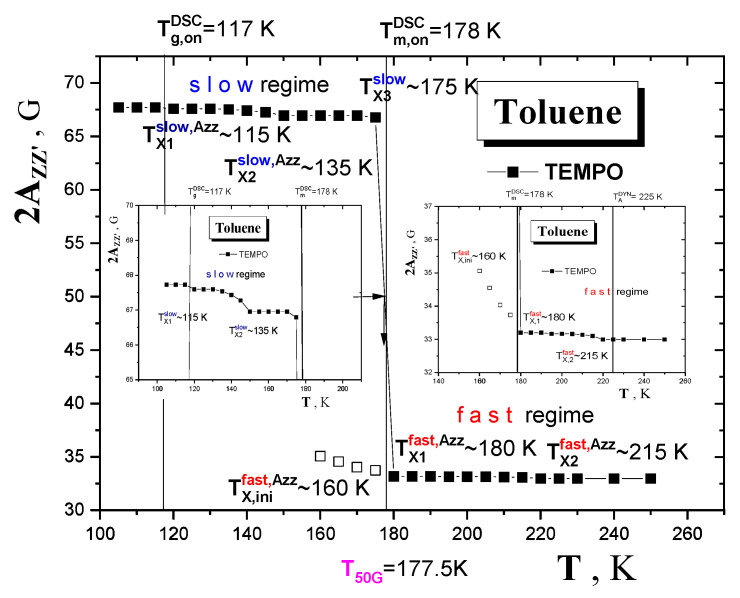
Spectral parameter of mobility 2*A*_zz‘_ as a function of temperature for the spin system TOL/TEMPO. Insets show the details of the slow and fast motion regime regions. Error bars are of the point size.

**Figure 8 materials-15-00462-f008:**
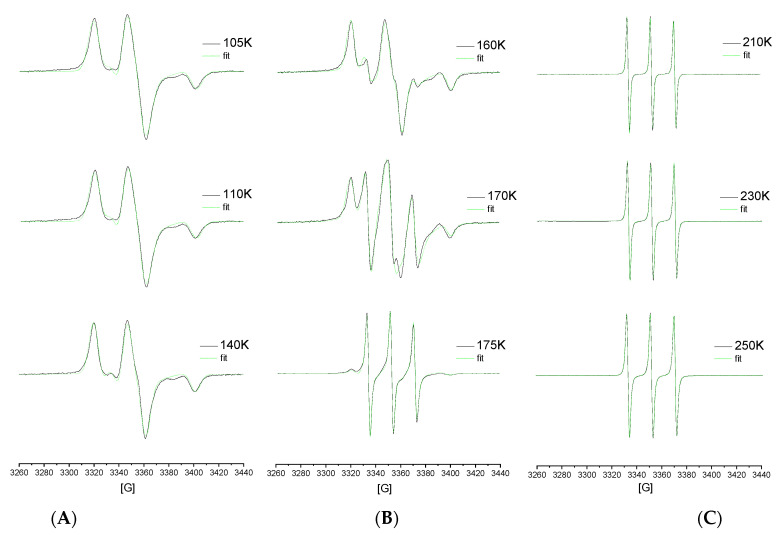
Experimental (black) and simulated (green) ESR spectra of the spin system TOL/TEMPO from three main regions of the two monomodal (**A**,**C**) and the one bimodal (**B**) type of ESR spectra.

**Figure 9 materials-15-00462-f009:**
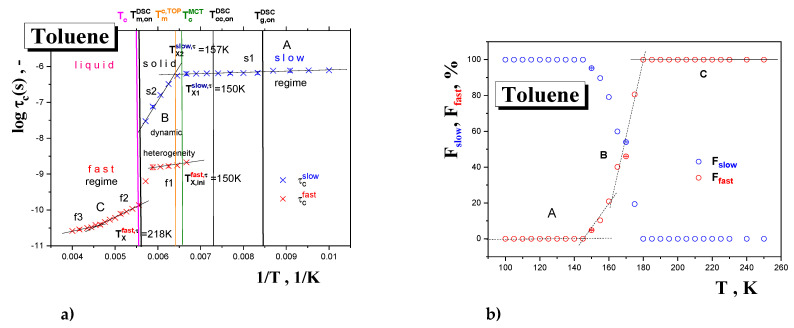
Arrhenius pot of the correlation time for the spin system TOL/TEMPO (**a**) and relative fraction of slow and fast spectral component as a function of temperature (**b**) as obtained via the NLSL program. Arrhenius equation fitting is as follows: slow region 1: *τ*_01_^slow^ = (3.5 ± 0.1) × 10^−7^ s, *E*_1_^slow^ = 0.35 ± 0.01 kJ/mol, *r* = 0.938; slow region 2: *τ*_02_^slow^ = (3.6 ± 0.05) × 10^−19^ s, *E*_2_^slow^ = 18.5 ± 0.01 kJ/mol, r = 0.995; fast region 1: *τ*_01_^fast^ = (1.6 ± 0.06) × 10^−10^ s, *E*_1_^fast^ = 1.6 ± 0.04 kJ/mol, *r* = 0.929; fast region 2: *τ*_02_^fast^ = (7.2 ± 0.02) × 10^−14^ s, *E*_2_^fast^ = 5.7 ± 0.13 kJ/mol, *r* = 0.998; fast region 3: *τ*_03_^fast^ = (2.9 ± 0.1) × 10^−12^ s, *E*_3_^fast^ = 2.3 ± 0.06 kJ/mol, *r* = 0.999.

**Figure 10 materials-15-00462-f010:**
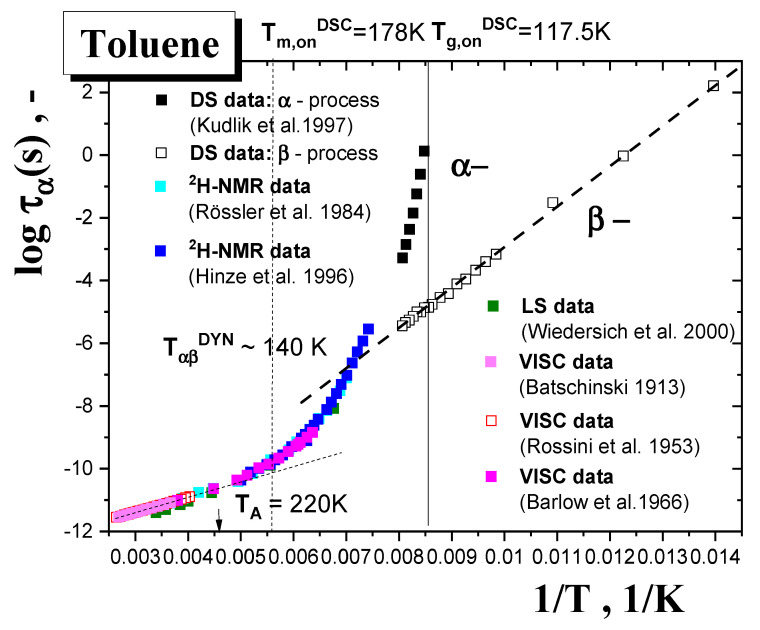
Relaxation map of the primary *α*− and secondary *β*−processes in amorphous TOL from DS [17], NMR [15], LS [18] and VISC [37,38,39] data.

**Figure 11 materials-15-00462-f011:**
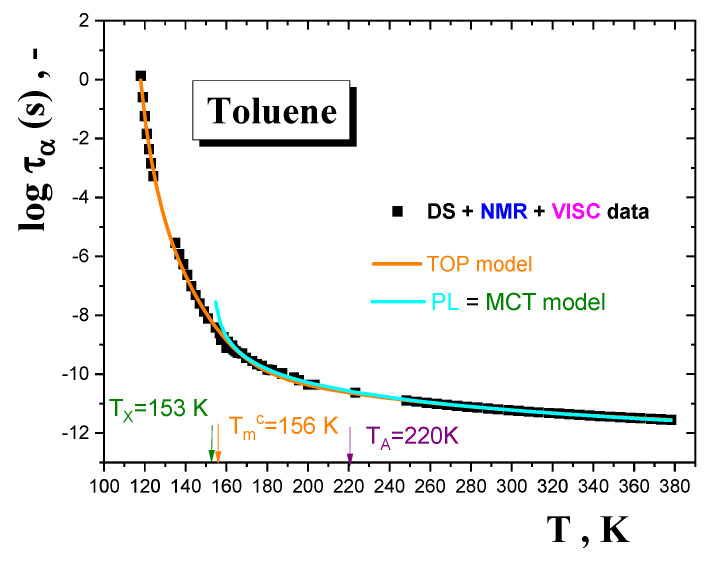
Fits of the structural *α* relaxation time in amorphous phase of TOL using the PL equation or the idealized MCT model and the TOP model. The PL or MCT parameters are: *τ*_∞,*α*_ = 5.6·10^−12^ s, *T*_X_^PL or MCT^ = 153 K and *µ* = 1.89. The TOP model parameters are: *τ*_∞,*α*_ = 1.5·10^−13^ s, *E*_∞_ = 4.6 kJ/mol, *T*_0_^TOP^ = 130 K, *T*_m_^c^ = 156 K and *T*_A_ = 220 K, taken from [55].

**Figure 12 materials-15-00462-f012:**
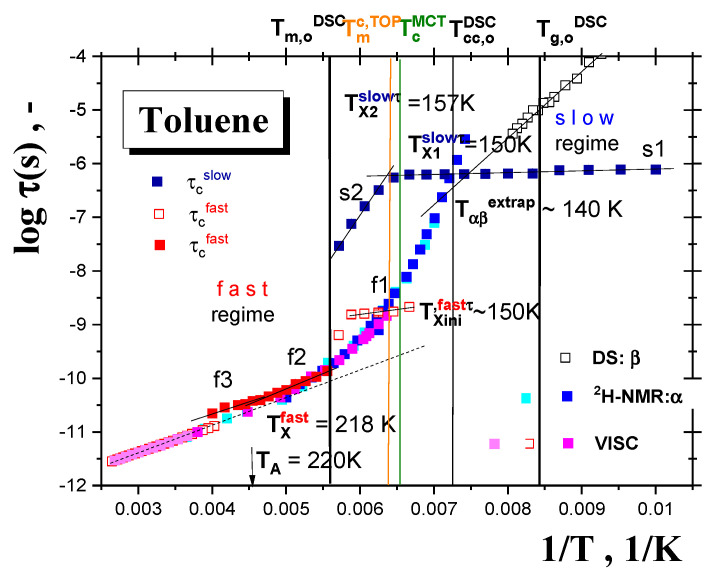
Comparison of the correlation times of TEMPO in the partially crystalline (below *T*_m_^DSC^) and amorphous (above *T*_m_^DSC^) spin probe TEMPO/TOL system with the time scales of the primary *α*– and secondary *β* relaxations in TOL from NMR [17,18,19,20] and VISC [51,52,53] data in the amorphous TOL medium.

## Data Availability

Data will be made available based on request to the authors.

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
