# Peer review of "A Combined Atomic and Molecular Probe Characterization of Aromatic Hydrocarbons via PALS and ESR: Methylbenzene"

_materials, 2022, doi:10.3390/ma15020462_

Round 1

Reviewer 1 Report

The article "A combined atomic and molecular probe characterization of aromatic hydrocarbons via PALS and ESR: Methylbenzene" is a complete and very harmonious work in which advanced analytical methods are applied. It has essentially supplemented the information obtained by traditional techniques. 

With respect to your work, I have to make only several small comments:

1) "Figure 1 is of very low quality. It is impossible to estimate the temperature range of the transitions because the scale on the axes is not seeable." 

2) Why some references have subdivisions within themselves (12, 15, 17, 18, 20, 23, 24, 26, 30, 35, 44, 49, 51). Is this really necessary?

Eventually, the article is written in good and clear language, well designed, and carried out at a high organizational and methodological level. The subject of the manuscript corresponds to the aims of the journal. Finally, the article can be published after some minor corrections.

Author Response

Dear Reviewer,

Thank you for reading our manuscript and giving a positive evaluation of the text.

Ad.1. It has rightly been noticed that the quality of the Figure 1 makes it difficult to read the temperature. The Figure 1 has been changed as suggested.

Ad.2. We have adapted the list of citation to the Reviewer's suggestions by removing subdivisions in Ref. 12, 15, 17, 18, 20, 23, 24, 26, 30, 35, 44, 49, 51. As a result of this change, the list of References now has 75 items, and the numbering of citations in the text has been changed.

Reviewer 2 Report

In this manuscript, the author reported a combined study of toluene via the PALS and ESR over a wide temperature range 300 K is reported, and the physical origins of the changes in free volume expansion and spin probe mobility are revealed. This is a meaningful work in the investigation of phase transition. The whole MS was in good organizing, therefore a minor revision of this manuscript is recommended.

  • There are some figure and captions should be revised. Figure 1, the DSC curve is ambiguous with low resolution, and the axis is unvisible. There is no mark of A,B,C in the Figure 6 and 8. The scale of abscissa in Figure 10 should be enlarge.

  • How much is the thickness of the chamber dedicated to liquid measurements? Does it affect the test result?

  • In the section of “Discussion”, in the discussion of “ the mutual relationship”and “4.2 thermodynamic interpretation”, there was no compared analysis with the literature.

  • There are some formatissues should be revised, such as “sometimes found to be coinciding.  This could suggest the same or at least similar origins”, “probe mobility or free volume, respectively. In a few cases by  comparisons of the”, “in the Figure 1 and for two „boundary" cooling regimes”

  • The language should be polished, there are many long sentence, which is hard to understand, such as “Finally, the physical origins of the changes in free volume expansion and spin probe mobility reflected in a series of the mutual coincidencies between the characteristic PALS and ESR temperatures are revealed using appropriate complementary thermodynamic and dynamic techniques. ”

Author Response

Dear Reviewer,

Thank you for reading our manuscript and giving a positive evaluation of the text.

Ad.1. We improved the figures as the Reviewer suggested. It has rightly been noticed that the quality of the Figure 1 makes it difficult to read the scale description – the Figure 1 has been changed. The additional description of A, B and C has been added to the Figures 6 and 8. The Figure 10, magnified.

Ad.2. Measurements in the solid and liquid phases were performed in the same measuring chamber, so there is no risk that the results will depend on the type of chamber used. The chamber resembles a small thermos flask, a vacuum is kept between the inner and outer walls for the purposes of isolation and thermal stabilization; the inner cylindrical copper vessel has an inner diameter of 11 mm and a wall thickness of 1 mm; the outer steel cylinder has a diameter of 23 mm and a wall thickness of 2 mm. Information in this regard has not been changed in the text.

Ad.3. The discussion chapter was divided into sub-chapters, in which the first two, indicated by the Reviewer, were intentionally distinguished as a discussion on the mutual correlation of the results obtained with the three techniques used in the work. In these sub-chapters, we found it unnecessary to re-refer to the previously cited literature (in the chapter Introduction). At the same time, we would like to note that the third sub-chapter very extensively and in detail indicates the reference to literature.

Ad.4. and 5. The indicated sentences have been changed - they are more precise in their current form.

Reviewer 3 Report

The current paper deals with the correlation between the annihilation parameters obtained from PALS and the rotation parameters revealed by ESR in the study of the mobility of TEMPO molecule in toluene. By means of DSC measurements, the principal transition events are characterized over a wide range of temperature, and then, related with the results obtained with PALS and ESR. Finally, the relation between the DSC and PALS and ESR parameters are discussed in terms of simple models based on liquid state dynamics.

The article is interesting and deserves publication in Materials's journal, since the obtained results shed light on the correspondence between different characterization techniques. However, prior to the acceptance, I would recommend to improve the article. On the one hand:

Regarding the english language and style, there are several sentences that are 5-6 lines long (e. g., the first sentence in the introduction, the 3rd sentence in the introduction, where in addition to its lenght, there is a missing verb). These sentences make reading a bit difficult and losing the thread of the article. Thus, I would recommend to re-check the article in order to avoid such long sentences.

Some suggestion/recommendations about the text, in general, are:

  • There are several parameters/acronyms that are not defined in the first place. I find this point quite crucial for the understanding of the article. Although the parameters/acronyms used may be trivial in their respective techniques/fields, they have to be defined; TXislow ....(see line 63), T50G,TcMCT,... Reading the article with so many undefined acronyms is a bit tiring. In addition, it would be nice if in the experimental section there would be a paragraph indicating how the characteristics temperatures are labelled for different characterization techniques and sample states (and listed). It seems that new acronyms appear while advancing in text, which for sure are labelled following a certain criteria, but the latter one is missing. In the current state, is really difficult to keep in mind the different samples' state, the different cooling and heating rates, the characteristic temperatures of each techniques mixed with different heating/cooling rates and samples' state... It is simply a mess.
  • Some misspells can be found over the article: (line 158: "K up to ca 200...", (line 216 "exthermic"). Check the article. 
  • Figure 1: The quality of the figure is not acceptable at least in the printed version. One can barely see numbers in X and Y axes. Moreover, the temperature is in ºC instead of K. Please update from ºC to K in order to be able to compare the temperatures with Figure 3, 4...
  • The label of the axes in all figures must be updated (fx, from "T, K" to "Temperature (K)", from \tau_3, ns to "tau_3 (ns)"...). 
  • The use of "i. e" is abusive. Break the sentences and start a new one when the sentences become too long when using "i.e".

Apart from the way the data is presented, there are few comments that I would like to address regarding the PALS experiments.

  • What is the lifetime value (or the range) related to the annihilation of free positrons? Authors can add this information to the experimental section for future studies and for the sake of reproducibility. 
  • In line 242, the authors say "(...) from the two PALS set up configuration". What do they mean with this sentence?
  • In lines 266-267, authors say that there is a step-like change in the o-Ps lifetime, shown in Fig. 3, which depends on the ΔT. What is ΔT? Heating-rate or heating step? And how the heating rate or step affect the o-Ps lifetime? Same goes for Fig. 5. The authors exhibit PALS results from measurements in a cooling mode with different temperature steps ΔT = -10, -5, -2. What do they mean by temperature-steps? The temperature interval at which PALS measurements have been measured? In this case what does the "ΔT = 10K and 5 K" label mean?
  • In lines 534-535 the authors state that in another organic compounds the measured o-Ps lifetime correspond to the local spherical free volume in TEMPO (about 2.72). In the case of the current paper, do the measured o-Ps lifetime agree (approximately) with the lifetime related with spherical open volumes at low temperatures? (∼1.3 ns) The Tau-Eldrup model could shed light on this issue.

Author Response

Dear Reviewer,

Thank you for reading our manuscript and giving a positive evaluation of the text. We have made effort to respond to the opinions of all Reviewers and to introduce corrections to our paper. Referring to the comments:

Ad.1. The article has been reviewed in terms of the sentences, and the long phrases have been simplified and divided into shorter sentences.

Ad.2. The indicated sentences have been reformulated. The parameters/acronyms used in the text are each time defined in introductory sentences, they correlate with the research technique and the type of transition within a given phase or phase transition. Additionally, the characteristic temperatures are indicated and signed in the figures.

Ad.3. Thank you for pointing out the typos - we've made the correction.

Ad.4. It has rightly been noticed that the quality of the Figure 1 makes it difficult to read the temperature.

The Figure 1 has been changed as suggested.

Ad.5. Both axis description formats are valid.

Ad.6. In a few sentences, the division was made in accordance with the Reviewer's suggestion, but in some sentences the use of i.e. is justified by the context - the sentences were left unchanged there..

Ad.7. As suggested by the Reviewer, the range of lifetime values related to the annihilation of free positrons has been added.

Ad.8. The measurement was performed in a helium cooling or nitrogen cooling configuration. Appropriate changes were introduced in the sentence indicated by the Reviewer and in experimental section.

Ad.9. The ΔT should be understood as the temperature interval between the measured PALS spectra. This step can also be read from the figures. In a sense, ΔT correlates with the sample heating / cooling temperature, but it should be remembered that the time of the sample storage at a given temperature is 1 hour (measurement time of one spectrum), and after that, another few minutes are needed to change the temperature for the next one.

Ad.10. In the article, we indicate the characteristic PALS temperature Tb1liq,t3 = 225 K (Figure 3), which correlates well with the characteristic ESR temperature TXfast,t = 218 K. The o-Ps lifetime measured at 218 K is 2.75(8) ns, and corresponds to the local spherical free volume in TEMPO (2.72 ns). This information was added to the text.